# Round-trip migration and energy budget of a breeding female humpback whale in the Northeast Atlantic

Lisa Elena Kettemer[1]*, Audun H. Rikardsen[1,2], Martin Biuw[3], Fredrik Broms[4], Evert Mul[2], Marie-Anne Blanchet[1,5]

**1** Faculty of Biosciences, Fisheries and Economics, UiT–The Arctic University of Norway, Tromsø, Norway, **2** Norwegian Institute for Nature Research, Tromsø, Norway, **3** FRAM—High North Research Centre for Climate and the Environment, IMR Institute of Marine Research, Tromsø, Norway, **4** North Norwegian Humpback Whale Catalogue (NNHWC), Straumsvegen, Kvaløya, Norway, **5** FRAM—High North Research Centre for Climate and the Environment, Norwegian Polar Institute, Tromsø, Norway

* lisa.e.kettemer@uit.no, le.kettemer@gmail.com

**Data Availability Statement:** Tracking data is available for viewing on www.movebank.org (ID

## Abstract

In the northern hemisphere, humpback whales (*Megaptera novaeangliae*) typically migrate between summer/autumn feeding grounds at high latitudes, and specific winter/spring breeding grounds at low latitudes. Northeast Atlantic (NEA) humpback whales for instance forage in the Barents Sea and breed either in the West Indies, or the Cape Verde Islands, undertaking the longest recorded mammalian migration (~ 9 000 km). However, in the past decade hundreds of individuals have been observed foraging on herring during the winter in fjord systems along the northern Norwegian coast, with unknown consequences to their migration phenology, breeding behavior and energy budgets. Here we present the first complete migration track (321 days, January 8th, 2019—December 6th, 2019) of a humpback whale, a pregnant female that was equipped with a satellite tag in northern Norway. We show that whales can use foraging grounds in the NEA (Barents Sea, coastal Norway, and Iceland) sequentially within the same migration cycle, foraging in the Barents Sea in summer/fall and in coastal Norway and Iceland in winter. The migration speed was fast (1.6 ms⁻¹), likely to account for the long migration distance (18 300 km) and long foraging season, but varied throughout the migration, presumably in response to the calf's needs after its birth. The energetic cost of this migration was higher than for individuals belonging to other populations. Our results indicate that large whales can modulate their migration speed to balance foraging opportunities with migration phenology, even for the longest migrations and under the added constraint of reproduction.

## Introduction

In many animal taxa, migration is a crucial behavior that allows organisms to match their life history requirements to environmental variability in space and time [1]. In many cases, feeding and breeding areas are geographically separated, sometimes by substantial distances. In

1064984327) and a reproducible dataset is uploaded as S1 and S2 Datas.

**Funding:** The project was financed by the Norwegian Research Council (Whalefeast project, RFFNORD no. 282469; https://www.forskningsradet.no) and LEK was funded by UiT – The Arctic University of Norway (https://uit.no). The funders had no role in study design, data collection and analysis, decision to publish, or preparation of the manuscript.

**Competing interests:** The authors have declared that no competing interests exist.

response, some long-range migrants have developed a capital breeding strategy, where they fast while in suitable breeding areas and accumulate energy reserves at suitable feeding grounds [2, 3]. The survival and reproductive success of capital breeders therefore relies on maximizing energy gain on feeding grounds and minimizing energy expenditure, to cover fasting periods of migration and reproduction. Because the amount of energy available to a capital breeder is fixed during the feeding period, capital breeders need to optimally match migrations to external conditions and internal state (e.g., prey availability and pregnancy, [3–5]). This requires careful budgeting of energy over the annual and life cycle [3, 4]. While there is currently no way to obtain continuous data on energy expenditure of free-ranging large cetaceans over their annual cycle, bioenergetic models can be used to estimate energy expenditure. Long-term individual tracking data provides information (such as movement speed) to parameterize these models [4, 6, 7].

Humpback whales (*Megaptera novaeangliae*) undertake the longest migrations of all mammals. In the northern hemisphere, they typically migrate from summer and fall feeding grounds at high latitudes to winter breeding grounds in specific tropical areas delineated by warm water temperatures [8, 9]. In the Northeast Atlantic (NEA), humpback whales forage in the Barents Sea and adjacent waters [10, 11], around Iceland [12] and Greenland [13], and then migrate to breeding grounds in the West Indies [14, 15] or the Cape Verde Islands [15, 16]. The distance between the Barents Sea and the West Indies represents the longest migration route of any humpback whale population (a great circle distance of ~ 9 000 km vs. 8 461 documented by [9]). Northwest Atlantic humpback whales also migrate to the West Indies from Newfoundland-Labrador, the Gulf of St. Lawrence, or the Gulf of Maine [15, 17–19], a distance up to 5 000 km. As a result of this long migration distance, NEA humpback whales may face high energetic constraints compared to whales migrating elsewhere. However, until now, no tracking data following NEA humpback whales on their migration has been available.

During the last decade, hundreds of humpback whales have been observed in specific fjord systems of northern Norway during winter (main season November–January [20–23]). Here, the whales feed on large aggregations of Norwegian Spring Spawning herring (*Clupea harengus*) that have overwintered in these fjord systems in this period [20, 22]. While within-season photographic matches have been reported between coastal Norway and the Cape Verde Islands [21], it remains unclear whether whales feeding in coastal Norway during the winter also migrate to breeding grounds in the West Indies within the same season. Given the traditional view of humpback whale migration phenology, feeding during the winter may present a shifted or extended feeding season, which may exacerbate the existing constraints of covering the long distance to breeding grounds and matching the timing of the reproductive season. However, since Norway lies between the Barents Sea and breeding grounds, feeding there may allow individuals to accumulate additional energy reserves prior to migration to increase breeding success [23]. While pregnant humpback whales commonly maximize the time spent on feeding grounds and leave later than other groups [24–26], this may present a critical trade-off if they must reach suitable waters prior to the end of gestation, for instance if early calf survival is influenced by water temperature or the availability of sheltered waters [9, 27].

Here, we present the first tracking data of a full round-trip migration for a humpback whale. The female was tagged at the feeding area in coastal northern Norway in January. Because the whale was observed without a calf when tagged, and then observed again with a calf upon its return to the same area in the following season, we had the unique opportunity to examine the round-trip migration covering late pregnancy, calving and lactation. We aimed to 1) describe the migration phenology and migration pathway through the NEA, 2) describe the movement characteristics of a female during pregnancy and lactation, and 3) estimate the energetic cost of this migration.

## Materials & methods

### Tag deployment and data pre-processing

We deployed a transdermal Argos satellite tag (SPOT-303, size: 300mm x 24mm, www. wildlifecomputers.com) on a female humpback whale on the winter feeding grounds in coastal northern Norway (Kvænangen fjord, January 8th, 2019, Fig 1). The tag was deployed from a 26-ft open rigid-hull inflatable boat using an Aerial Rocket Tag System launcher (LKARTS- Norway) from about eight meters distance. Tagging procedures were approved by the Norwegian Food Safety Authorities (Mattilsynet) under permit FOTS-ID 14135. We programmed the tag to send 16 transmissions/hour for the first 100 days, then 14 transmissions/hour for the following 30 days and then 12 transmissions/hour for the next 90 days. After 220 days, the tag sent 80 transmissions per day for the rest of the deployment. The sighting history of this individual was extracted from the North Norwegian Humpback Whale Catalogue (NNHWC, [28]) by matching photographs of its fluke (sighting history in S1 Table in S1 File). Sightings on the 2nd and 14th November 2019 confirmed the presence of a calf (Fig 2). All numerical and statistical analyses were performed using R software version 4.0.3 [29].

Because migration speed was one of the key parameters used in our analyses, the raw Argos locations were projected to an azimuthal equidistant projection centered on the middle of the track (45˚N, 20˚W) to best conserve distances between locations across the latitudinal range. Extreme outlier positions were then removed using a speed, angle and distance filter (max speed: 9 ms$^{-1}$, sda() function of the trip package, version 1.7.1, [32] based on [33]). All positions with quality class Z were removed. The filtered locations were then used to reconstruct the most likely path using a continuous time state-space model from the foieGras package using fit_ssm() in version 0.6–9 [34–36]. This model assumes an underlying correlated random walk process considers the error ellipse estimates around the original locations provided by CLS-Argos (S1 Fig in S1 File), and the most likely movement path and its associated uncertainty estimates are returned. We sampled locations along the predicted path at 6-hourly intervals, which were then used throughout our further analyses.

We also calculated the movement persistence, $\gamma$, between successive locations, which is the autocorrelation in speed and directionality for each step [35], using fit_mpm() in foieGras. Movement persistence characterizes a continuous behavioral mode, ranging from meandering movements associated with area restricted search behaviors (ARS, $\gamma = 0$) to directed movement ($\gamma = 1$) usually associated with transit. Switches in $\gamma$ therefore indicate changes in behavioral mode, e.g., between migration and residence. To identify start and end dates of migration, we conducted a segmentation analysis on $\gamma$, using the "Lavielle" method, lavielle() within adehabitatLT, version 0.3.25 [37–39].

To identify the calving date, we carried out the same analysis using speed as the parameter. Based on gestation duration, the whale would have given birth in the first months after leaving the feeding grounds, so we limited this analysis to the southbound leg of the migration. We assumed two different behavioral states (pre- and post-birth) distinguished by the difference in swim speed between pregnancy and lactation (i.e., accompanying a neonate calf).

To assess whether the whale deviated from the shortest possible path to the breeding grounds, we calculated the least cost path through water between the mean latitude and longitude of feeding and breeding locations, constraining the path to water depth > 10m (obtained from [30]), using the lc.dist() function in marmap, version 1.0.4 [40]. Two resulting alternative shortest distances are given between the feeding ground and breeding ground: either directly or via the observed stopovers (Fig 1).

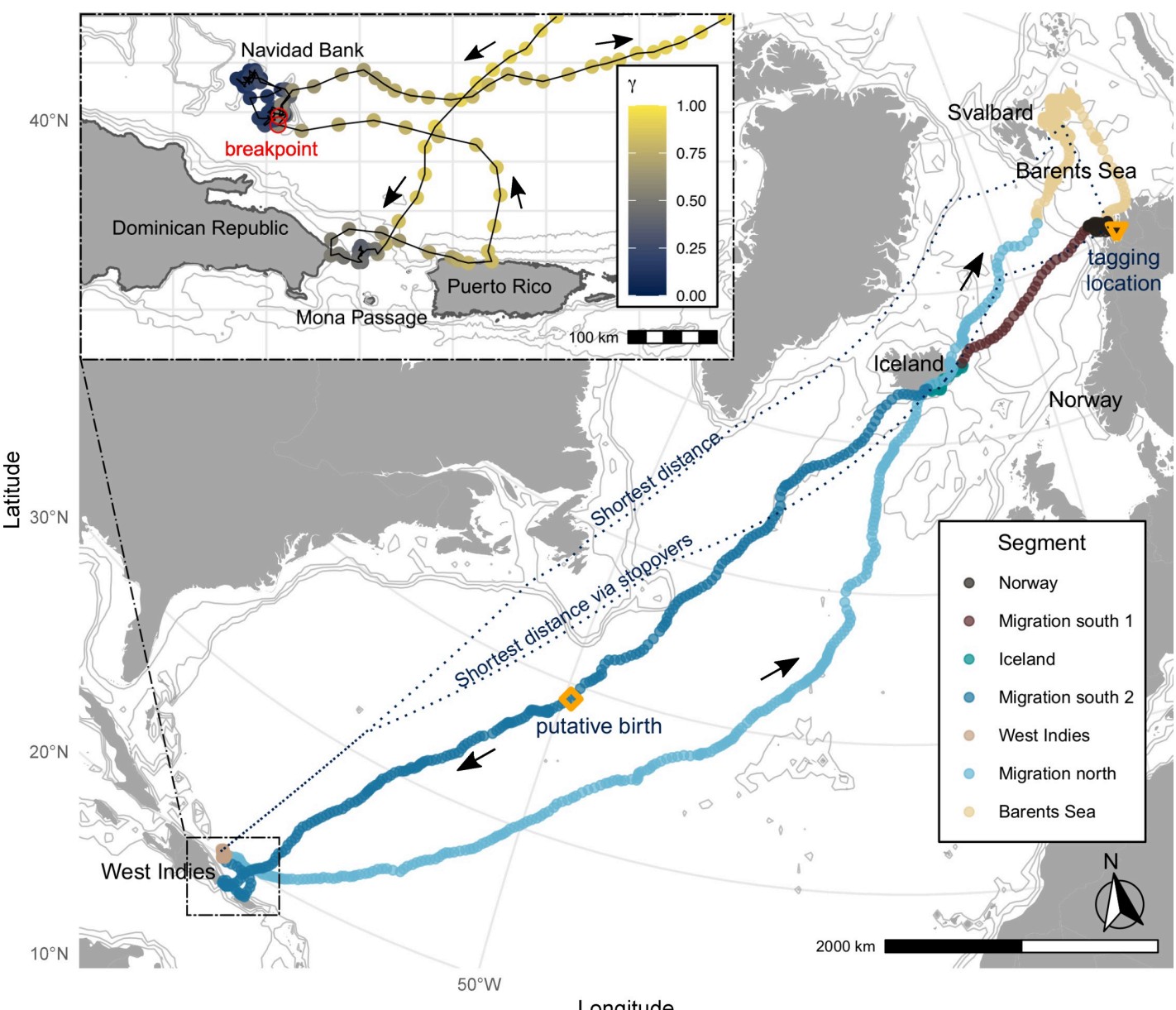

**Fig 1. Full migration track of a pregnant female humpback whale in the Northeastern Atlantic.** Locations from predicted path colored by the different migration segments. Tagging location is indicated by an orange triangle and the putative location of birth is indicated by an orange square. The inset shows movements on the breeding grounds, colored by movement persistence, yellow indicating directional travel, and blue indicating meandering movement. Arrows show the direction of movement. Dotted blue lines in the main map indicate the shortest possible distances through water. Contour lines show the 200 m, 1000 m, and 2000 m bathymetric isolines (data from [30] for the main map and [31] for the inset). Landmass data was obtained from naturalearth.

### Energetic model

We estimated the energetic cost of this migration using a bioenergetic model that estimates the energy required for an animal to cover 1) its basic energy requirements (basal metabolic rate, BMR) and 2) the cost of overcoming the drag forces associated with moving its body through water at a given speed (cost of transport, $E_{COT}$, based on [41]). The model parameters (Table 1) are described in [6] and [7]. We estimated BMR using the Kleiber allometric

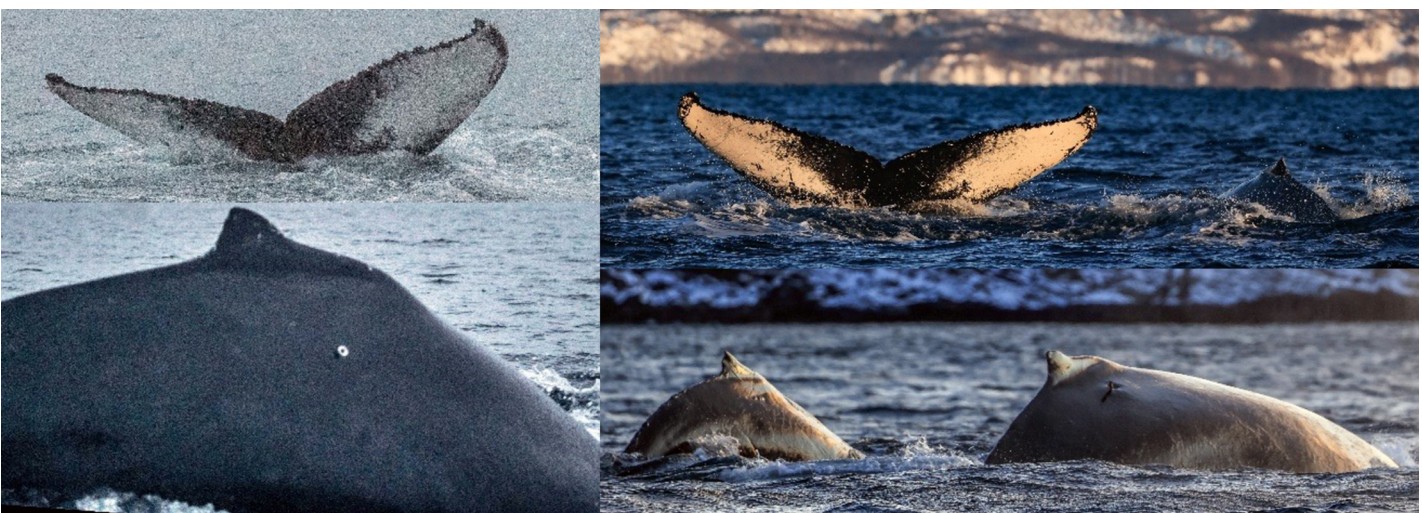

**Fig 2. Photographs from deployment and resighting with calf.** Left: Fluke and tag position at time of deployment (January 8[th], 2019). Right: First re-sighting the following season, the individual accompanied by calf (November 2[nd], 2019) with the tag still in place. Photo: A. Rikardsen.

equation of BMR in relation to body mass in kcal day[-1] [42]:

$$BMR = 4186.8\,[70M^{0.75}] \qquad \text{(Eq 1)}$$

where $M$ is body mass (kg). We assumed a body mass of 30 000 kg for a 13 m female (in accordance with [6, 43]). We multiply by 4186.8 to convert to Joule day[-1] and then multiply by the migration duration (in days) which gives the metabolic maintenance cost (Joule).

The energetic cost of transport $E_{COT}$ is calculated as:

$$E_{COT} = \left(\frac{\lambda}{2\varepsilon_A\varepsilon_P}\right)\rho S C_d V^3 \qquad \text{(Eq 2)}$$

Here, the aerobic efficiency $\varepsilon_A$ describes how efficiently metabolic energy is converted to mechanical work by the muscles, and $\varepsilon_P$ is the propulsive efficiency describing how efficiently mechanical work is converted to forward motion [44]. The $C_d$ term is the drag coefficient, $\rho$ is the density of water (kg m[-3]) and $S$ is the wetted surface area of the whale (m[2]). The ratio of active to passive drag, $\lambda$, accounts for the fact that active body movements and posture changes change how drag forces act on a body moving through a medium [41]. Finally, $V$ is the speed of the animal through water (ms[-1]), estimated from displacement between predicted locations.

**Table 1. Parameters, units and values used to calculate the energetic cost of migration.** Parameter estimates were chosen in accordance with [6, 7].

| PARAMETER | UNIT | VALUE | SOURCE |
|---|---|---|---|
| $C_D$ | Drag coefficient | 0.003 | [6] |
| S | Wetted surface (m[2]) | 0.054 M[0.696] | [45] |
| M | Mass (kg) | 30 000 | [43] |
| $E_P$ | Propulsive efficiency | 0.8 | [46] |
| $E_A$ | Aerobic efficiency | 0.2 | [46] |
| $\lambda$ | Ratio of active to passive drag | 0.7 | [41] |
| P | Density of seawater (kg m[-3]) | 1 027 | Standard for seawater |
| V | Swim speed (ms[-1]) | dynamic | Displacement between locations, constant for each 6-hour step |

Eq (2) gives the instantaneous power required to overcome drag (in Watt, or Joule s$^{-1}$) at a given speed. Our data consists of interpolated positions at 6-hour intervals, providing swim speed estimates for each of these intervals. To convert the instantaneous power (Eq 2) to the energy expenditure required to swim at this speed for the duration of each interval, we therefore multiply it by the timestep duration (i.e., 6 hours). To obtain an estimate of the daily cost of transport, we then sum every set of four 6-hour estimates. We then calculated the total energetic cost of migration as the sum of estimated metabolic maintenance cost and cost of transport ($E_{COT}$), for the duration of the round-trip (139 travelling days, 14 days along Iceland and 17 days on breeding grounds).

Because we use swim speed to estimate the energy requirements of migration it is important to account for the way ocean currents may assist or impede movements, i.e., how much of the observed displacement is due to movement by the whale itself. We estimated speed over ground from the geodesic distance between consecutive predicted locations using geodist() in the geodist package [37]. We then corrected speed over ground for ocean surface currents to obtain estimated swim speed through water. We used the nearest available record of surface current from a coupled atmosphere-land-ocean-sea ice model at quarter-degree spatial and hourly temporal resolution [47] (S1 Fig in S1 File). The ocean current data were prepared by the UK Met Office, Exeter, UK and made available online by E.U. Copernicus Marine Service Information [48]. We extracted the *u* and *v* current vectors nearest to each observed whale location in time and space. We followed [49] using wind-Support() in the package windR [50] (accessed 15.02.2021). Speeds presented throughout the manuscript refer to speed through water (current- corrected swim speeds). We classified each location during the migration as either resting ($< = 0.5 \text{ms}^{-1}$) or transiting ($> 0.5 \text{ms}^{-1}$ in accordance with [6]). Summary statistics are given as medians and 25$^{th}$ and 75$^{th}$ quantiles ($Q_{25}$-$Q_{75}$), unless otherwise stated.

## Results

We satellite tracked a female humpback whale for 321 days (January 8$^{th}$, 2019 – December 6$^{th}$, 2019) from the fjords of northern Norway to the West Indies and back to the same Norwegian fjord (November 2019), via the Barents Sea (Fig 1). The whale was last photographed in the same fjord on January 2$^{nd}$, 2020. Previous photographic records from the North Norwegian Humpback Whale Catalogue (NNHWC) show that it was present on the Norwegian coastal feeding grounds also during the 2013/14 and 2014/15 winter seasons (NNHWC, S1 Table in S1 File). We identified seven segments in the track which corresponded to changes in the movement persistence mean: 1. Norway, 2. Transit to Iceland ("migration south 1"), 3. Iceland, 4. Transit to breeding grounds ("migration south 2"), 5. West Indies, 6. Transit to feeding grounds ("migration north"), 7. Barents Sea and Norway (Fig 1). The breakpoints between segments identified departure and arrival times (dashed lines Fig 3).

The whale left the Norwegian fjords on February 7$^{th}$, 2019, to the east coast of Iceland, where it spent 15 days before continuing the southward migration on March 1$^{st}$ (Fig 3). It was present on known breeding grounds from mid-April to mid-May. After its return to the NEA, the feeding season lasted between late July and late October 2019 in the Barents Sea, and between early November and at least until the last sighting on January 6$^{th}$, 2020, in the Norwegian fjords ($> 5$ months). We found that using the stopovers in Norway and Iceland increased the shortest possible distance between feeding and breeding grounds by 7.5% to 9 071 km (Fig 1). The cumulative distance of the observed migration path was 18 500 km, split between 9500 km on the 68-day southward migration (coastal Norway–West Indies) and 9000 km on the 71-day northward migration (West Indies–Barents Sea). The whale deviated from the shortest

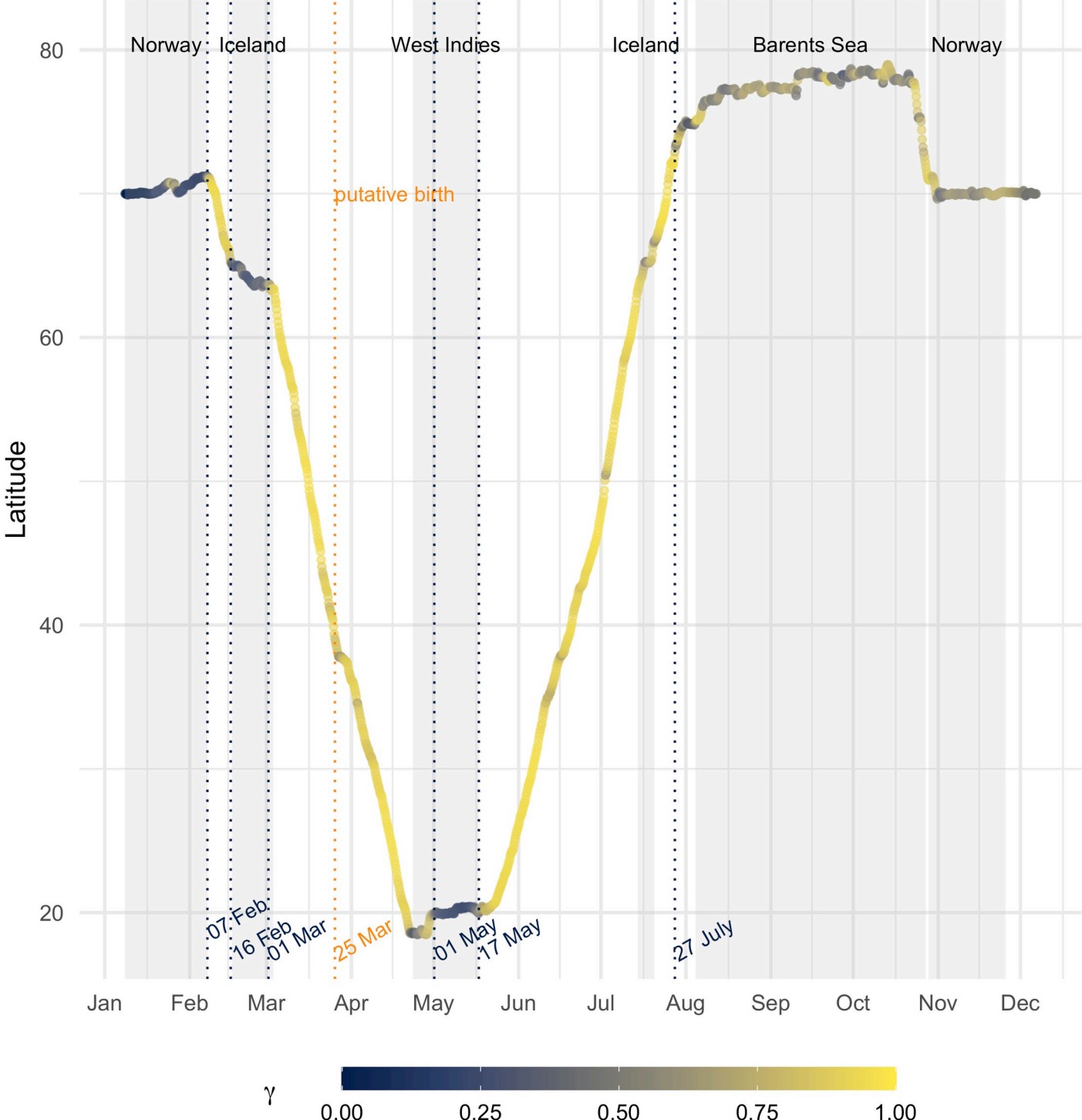

**Fig 3. Migration and movement persistence.** Migration of the whale as indicated by the change of latitude over time with movement persistence, $\gamma$, at the respective location shown in color (0–1). Background shading indicates when the whale was observed over the continental shelf (water depth $<$ 200 m [32]). The time spent on shelf regions are associated with low $\gamma$ in dark blue indicative of area restricted search behavior, while high $\gamma$ in yellow indicates transit. The blue dotted lines indicate the break points based on changes in mean movement persistence.

path and followed different paths during the southward and northward migration, with the northward path further to the east (Fig 1).

Based on the segmentation, we define migration as the part of the track between coastal Norway until switching to low movement persistence south of the Barents Sea. During this migration, 139 days were classified as transiting (speed: 1.5 ms$^{-1}$, Q$_{25}$-Q$_{75}$: 1.1–1.8) and 31 days as stationary at stopover areas in Iceland and on the breeding ground in the West Indies (Fig 3). Migration speed was fastest in the first migration segment ("migration south 1", Norway to Iceland: 1.9 ms$^{-1}$, Q$_{25}$-Q$_{75}$: 1.7–2.2, Fig 4). We detected a breakpoint, i.e., a change in mean speed, during the second migration segment ("migration south 2") from Iceland to the West Indies. This was associated with a shift in median speed from 1.7 ms$^{-1}$ (Q$_{25}$-Q$_{75}$: 1.4–2.1) to 1.3 ms$^{-1}$ (Q$_{25}$-Q$_{75}$: 0.8–1.6, Fig 4). After an initial sharp decline and short period (24 hours) of very slow speeds ($< 0.5$ ms$^{-1}$), speeds increased again but remained lower than during the early migration (Fig 4). This suggests that calving occurred just after crossing the Gulf Stream/North Atlantic Current at 39˚N 49˚W, on March 25$^{th}$ (Fig 1) in sea surface temperature around 18˚C. Most resting ($< = 0.5$ m/s) occurred after this date (Fig 4, red circles). During the northward migration median speed was 1.5 ms$^{-1}$ (Q$_{25}$-Q$_{75}$: 1.1–1.8). Therefore, median speed prior to the putative calving date was overall faster (pre-calving: 1.8 ms$^{-1}$, Q$_{25}$-Q$_{75}$: 1.5–

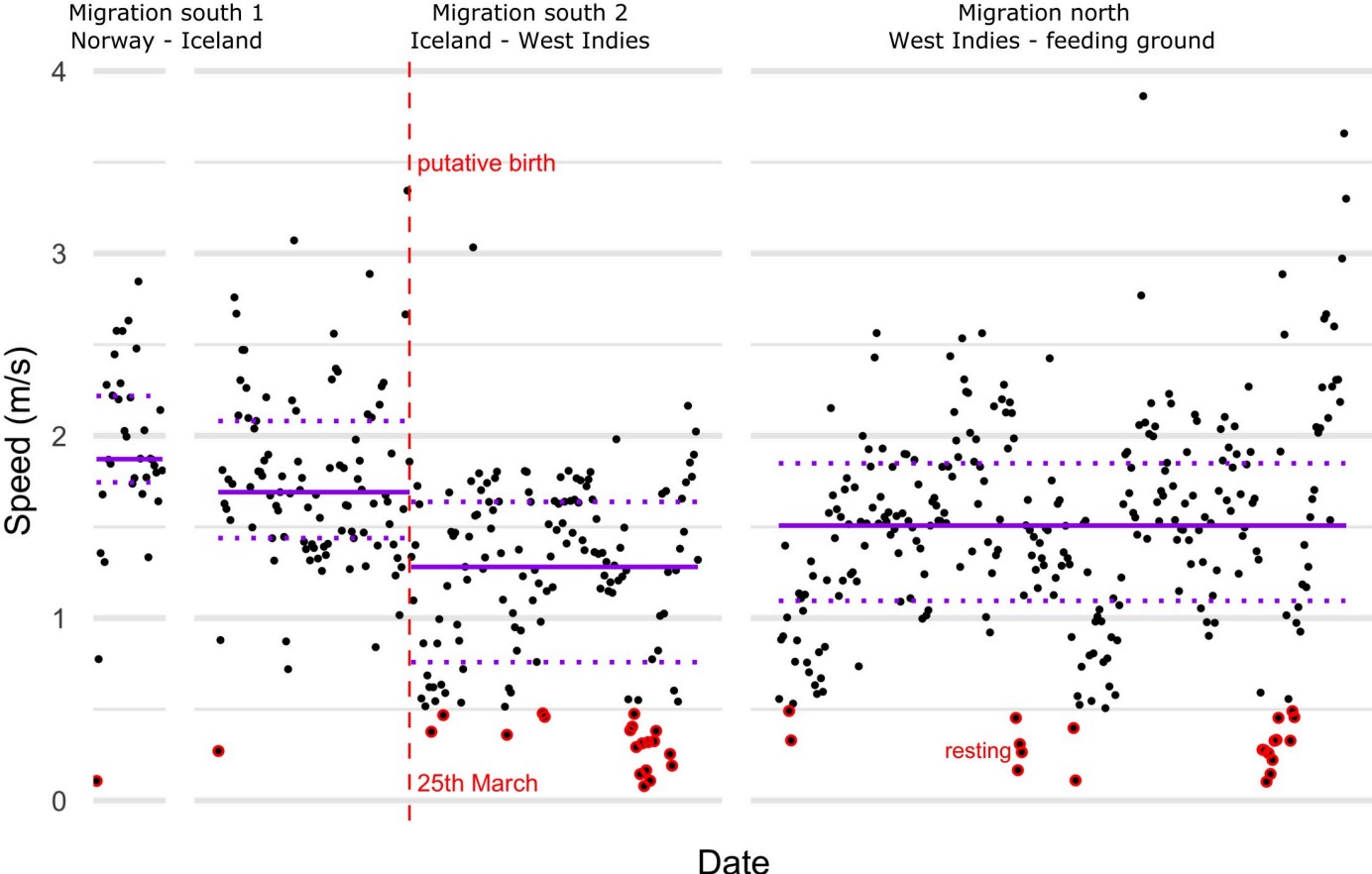

**Fig 4. Speed during the migration segments.** Chronological from left to right: Norway–Iceland, Iceland–West Indies, West Indies–Barents Sea. Segmentation based on movement persistence. The dotted red line indicates the putative time of calving based on a segmentation analysis of the "southward migration 2" segment, which was associated with a change in median speed from 1.7 ms$^{-1}$ to 1.3 ms$^{-1}$ (horizontal lines indicate median speed and dotted lines the respective first and third quartiles).

2.1) than when the whale was presumably traveling with her calf (post-calving: 1.4 ms$^{-1}$, $Q_{25}$-$Q_{75}$: 1.0–1.7, Fig 4).

The mother-calf pair reached Mona Passage (between Puerto Rico and Hispanola) on April 21$^{st}$ and then moved between coastal areas of the Dominican Republic and the north coast of Puerto Rico until April 28$^{th}$ (Fig 1). For ~ 3 days, the pair seemed to be resting (speed < 0.5 ms$^{-1}$) but the end of southward migration (i.e., a breakpoint associated with a shift in mean movement persistence) was only detected after the pair transited to Navidad Bank on May 1$^{st}$ (Figs 1 and 3). They remained there until initiating the northward migration on May 17$^{th}$. On the northward migration, the mother-calf pair passed close to the Iceland coast but continued moving north. The end of migration was detected on July 27$^{th}$, south of the Barents Sea (73°N, 9°E, Figs 1 and 3). After ~3 months they proceeded towards the Norwegian coast (October 23$^{rd}$), where they arrived in early November.

Using the bioenergetic model, we estimated the total cost of the round-trip migration (from coastal Norway to the Barents Sea) to be 142 030 megajoule (MJ), 28 360 MJ allocated to cost of transport and 113 670 MJ to maintenance metabolism (Table 2). This total estimate includes energetic expenditure for 170 days (coastal Norway–Barents Sea), of which 139 days were spent actively transiting ("migration south 1", "migration south 2", "migration north") and 31 days stationary on stopovers in Iceland (14 days) and on the breeding ground in Navidad Bank (17 days). We also present a theoretical cost at lower migration speeds (as reported in previous studies, 0.9 m s$^{-1}$ and 1.1 m s$^{-1}$) in S2 Table in S1 File. The whale was likely lactating for 124 out of the 170 days, given the putative calving date.

## Discussion

We present the longest and first fully recorded round-trip migration of a humpback whale, and the first satellite track of an individual from the Northeast Atlantic (NEA). While earlier studies have provided key insights into humpback whale movement and migrations in many other regions, these studies have only mapped migrations one way, due to the limited longevity of tag deployments. Most studies tracked humpback whales from breeding grounds and in one published case from feeding grounds [7, 19, 51–53]. We show connectivity between three known NEA feeding grounds, describe the phenology of this migration, and provide new detailed information on the movements of a mother-calf pair. We were also able to estimate

**Table 2. Energetic cost of migration.**

| PARAMETER | A) FULL MIGRATION INCL. ICELAND AND BREEDING GROUNDS | B) NORTHWARD MIGRATION |
|---|---:|---:|
| **DURATION** (days) | 170 (transiting: 139) | 71 |
| **DISTANCE** (km) | 18 500 | 9 000 |
| **MEDIAN SWIM SPEED** (ms$^{-1}$± IQR) | 1.3 ± 1.2 (transiting: 1.6 ± 0.7) | 1.5 ± 0.8 |
| **E DAY$^{-1}$** (MJ Day$^{-1}$) | 834 | 867 |
| **E$_{COT}$** (MJ) | 28 360 | 14 309 |
| **METABOLIC MAINTENANCE** (MJ) | 113 670 | 47 738 |
| **E$_{TOTAL}$** (MJ) | 142 030 | 62 047 |

Energetic cost of migration for **A)** a full migration for a female traveling with a calf from Norway to West Indies, including a stopover in Iceland, **B)** values for the northward migration, from West Indies breeding grounds to the start of foraging behavior south of Barents Sea feeding grounds. Values presented are migration days, kilometers traveled (km, as measured along the track), the energetic cost of transport (E$_{COT}$), the cost per day, metabolic maintenance (total BMR expenditure) and the total energetic cost (sum of E$_{COT}$ and metabolic maintenance). Energetic costs are presented in megajoule (MJ).

the most likely location and timing of calving during the southward migration and show how this affected the whale's migration speed. This track provided unique empirical data to estimate the energetic requirements of this migration route. It therefore allowed us to explore the trade-offs associated with allocating time between feeding, breeding, and migration throughout the annual cycle.

We demonstrate that humpback whales can use several NEA feeding grounds (the Barents Sea, coastal northern Norway, Iceland) and migrate to the West Indies to give birth within the same annual cycle. The connectivity between these feeding grounds in the NEA has been sporadically documented by photographic matches between Iceland and the Norwegian coast [54], between the Barents Sea and Iceland [Broms, pers. communication], and between the Barents Sea and the Norwegian coast [23]. Sequential use of these three areas within one annual cycle had not been recorded prior to this study. The tracked individual foraged in northern Norway until early February, and likely for two additional weeks in Iceland [12]. Humpback whale winter feeding aggregations have previously been described from Iceland [55] and Alaska [56], and at least some of the observed individuals migrate at the end of this feeding season. Only a fraction of the whales feeding in the Barents Sea are observed at the coast of northern Norway in the winter [11, 23]. Many of these have been observed in several seasons, suggesting there may be an individual preference for this feeding strategy [23]. The remainder of the population may travel directly from the Barents Sea to breeding grounds, or continue feeding offshore where they are less easily observed, as there is also high herring biomass offshore in the winters.

The whale left Norway on February 7th, slightly later than other whales, as most leave between December and late January [23]. In other regions, pregnant females also remain up to two months longer on the feeding grounds compared to other groups [24–26], presumably to cover the substantial additional cost of pregnancy and lactation [57, 58]. Females can maximize their calf's chances of survival by providing sufficient energy and resources both during pregnancy and subsequent lactation [59, 60]. This strategy may increase reproductive success of the mother if it is successfully employed across years [24, 57, 61].

The female appeared to be able to partly compensate for the late departure from feeding grounds by increasing its overall travel speed during transit, thus arriving towards the end of, but still within, the breeding season of NEA humpback whales [15]. Whales from the NEA exhibit a later breeding season in the West Indies (February–May) than whales from the Northwest Atlantic. This may be due to their longer migratory distance [15, 62], or different seasonality of feeding. Based on this difference in breeding season and their spatial use of the West Indies, NEA humpback whales might form a behaviorally distinct population segment [15]. Most NEA humpback whales are sighted in the east of the West Indies [15], but the tracked individual frequented areas predominantly associated with Northwest Atlantic humpback whales further west (Dominican Republic, incl. Navidad Bank). Whale presence in the Dominican Republic peaks in February and early March, with few sightings in April [63], but our data substantiates recent evidence from acoustic monitoring that the season in this area lasts until the end of May [64].

Calving occurred during the southward migration, ~35 days before the female reached the breeding grounds. Calving is generally expected in calm, shallow coastal or bank waters [65–69] and not in exposed oceanic waters. However, the observed changes in movement (i.e., from fast to very slow movement followed by a period of continuous movement at reduced speed) are consistent with the behavior reported from earlier observations of calving events [65, 66, 70]. Newborn calves have been documented outside of the described main breeding grounds elsewhere [26, 71] and historical whaling records from Norway include records of late-stage pregnancies in Norwegian waters during winter and spring [72], also indicating

humpback whales from this region might give birth shortly after these observations were made, likely outside breeding grounds [73]. This indicates that shallow waters are not crucial for neonates immediately after birth, but that perhaps water temperatures may need to be above some critical value [9]. It may also reflect that maximizing maternal energy intake in productive waters is more important to calf survival than a birth in shallow, warm breeding grounds.

The overall migration speed (1.5 ms$^{-1}$) was faster than documented for humpback whale migrations in most other regions and varied throughout the migration. Speed was highest during pregnancy early in the migration (1.7–1.9 ms$^{-1}$) and remained relatively high during nursing (1.4 ms$^{-1}$). Previously reported average migration speeds obtained from satellite tracks ranged between 0.9 ms$^{-1}$ [7] and 1.63 ms$^{-1}$ [52], the fastest being reported from the only other available tracking study covering the migration from feeding to breeding grounds [52]. Whales satellite tagged on the West Indies breeding ground and heading towards the NEA were also slightly faster compared to those heading to western feeding grounds, presumably due to their longer migration distance (1.25 ms$^{-1}$ [19]). None of these studies accounted for ocean currents, but speed was overall only slightly influenced by currents (mean absolute difference between speed through water and speed over ground 0.07 ms$^{-1}$, S2 Fig in S1 File). The fast migration speed may be a response to time constraints in reaching the breeding grounds given 1) the long distance between Norway and the West Indies, 2) the late departure from Norway at the end of the winter and 3) the additional stopover in Iceland. Elephant seals for example also increase migration speed to precisely time their arrival to breeding grounds where they give birth [5], and our data indicate this may be the case in humpback whales.

While fast migration speeds may be expected for NEA humpback whales, nursing whales are generally expected to migrate slower than those traveling without calves (0.86–1 ms$^{-1}$ [6, 68, 74]). Although the present female adjusted her speed when traveling with the calf, the migration speed still exceeded previous estimates of theoretical optimal migration speed estimated for nursing humpback whales for similarly long migrations (1.1 ms$^{-1}$ for ~ 8 500 km one-way [6]). After the putative calving, swim speeds decreased, and more days were spent resting. However, we identified fewer resting days compared to what has been previously reported as optimal for energy conservation, calf growth and milk transfer rates (14% compared to 27% [6]). Routine swim speeds for baleen whales, i.e., the speed at which animals swim most efficiently based on physical adaptations, seem to converge around 2 ms$^{-1}$ during transiting movements on feeding grounds [75, 76]. Swim speeds slower than 2 ms$^{-1}$ observed from tracking data may therefore be due to resting periods between swimming bouts not resolved at 6-hourly resolution.

In other regions, lactating humpback whale females choose migration routes close to coastlines and spend time in sheltered areas to rest and nurse [68, 77]. In contrast, the migration of NEA whales occurs almost entirely on the high seas. Therefore, mother-calf pairs may have different resting/energy conservation strategies. Reduced resting time and faster swim speeds can lead to a loss of milk and increased energetic demands of the calf, increasing the energetic cost to the mother beyond that solely caused by higher costs of transport at higher swim speed [6]. Because we cannot reliably quantify this effect (but see [6]), we did not include the cost of lactation and gestation. Therefore, the total energy expended by the mother during migration will be higher than reported.

The fast migration speed and long distance resulted in a high energetic cost of this migration, compared to e.g., lactating females migrating 0.6 ms$^{-1}$ slower and ~ half the distance from Australia to the Southern Ocean [7]. However, the fast migration speed reduced the time required to complete the distance (70 compared to a mean of 62 days reported by [7]). This means that some of the additional cost caused by fast speeds was offset by reducing

maintenance costs, compared to if the migration duration had been extended by slower swim speed. If the whale in this study had migrated at a slower speed during the northward migration, e.g., the theoretical optimal migration speed reported by [6] (1.1 ms$^{-1}$) or the average swim speed of mother-calf pairs reported by [7] (0.9 ms$^{-1}$), its arrival to the feeding ground would have been delayed (24 or 45 days, respectively). While reducing the cost of transport, this would have caused the whale to miss part of the feeding season and would have incurred higher maintenance costs for each additional day of migration. Migration strategies available to whales feeding throughout the winter in Norway may therefore be to 1) travel fast, spend time on breeding grounds (as we observe), 2) travel slowly and reduce time on breeding grounds or 3) travel slowly and start feeding later in the year (e.g., mid-August/mid-September).

Obtaining reliable measurements of biomechanical parameters and travel speeds is challenging for large marine vertebrates, so bioenergetic models such as the one presented here, rely on a range of assumptions and approximations [6, 78]. While parameter values in the model are averages of estimates, true values are likely variable between individuals and over time (due to differences in mass, condition, surface area, gaits, appendage morphometrics, behavioral patterns). For example, we use a fixed value for the drag coefficient ($C_d$) which has been estimated by previous studies. Substantial uncertainty exists around this value for most species, including humpback whales ([6] but see [76] and [79]), and it also varies with many of the same factors mentioned above. Similarly, BMR cannot be directly measured for large free-ranging marine animals, and there is disagreement on the relationship between size and metabolic cost for large animals [80, 81], so we rely on estimates based on allometry. Since both $C_d$ and BMR are central parameters in this model, our estimates are only rough indicators. Importantly, changing the values for BMR and $C_d$ in the bioenergetic model may lead to a shift in the relative importance of maintenance cost vs. transport cost, i.e., the importance for a whale to minimize metabolic cost (by decreasing the duration of migration) vs. minimizing transport cost incurred from movement (by decreasing speed). By using the same parameters as previous studies, we can ascertain that the relative cost of this migration is larger than that reported previously [7], but the magnitude of the increase cannot be determined with certainty.

Additional uncertainties exist regarding the energetic costs of mother-calf pair movement and how these scale with speed, since calves swim directly at their mothers' side, thereby changing the mothers' drag profile and their own [82]. While optimal swim speeds seem to be largely independent of size across the range examined in a recent study (minke whales—blue whales [75]), calves have less muscular power and lower lung capacity than adults [83]. Therefore, calf requirements and swim speed likely determine resting periods and overall migration speed. Furthermore, animals likely swim at a depth where additional wave drag produced at the air/water interface is minimized (e.g., ~12 m for large baleen whales) [84]. Ocean current speed at depth may differ to current speed at the surface. As the tag used in this study did not collect dive information, we assumed that the whale swims at an optimized depth most of the time, avoiding additional wave effects at the surface, and that the ocean current effects experienced can be approximated by surface currents (0–5 meters depth).

The behavioral choice observed in this study (i.e., longer feeding season and faster migration speed), and the resulting higher energetic cost, indicates a trade-off between benefits incurred from spending time on the breeding grounds with the calf (e.g., optimized nursing and growth, predator avoidance summarized in [27, 67]) and resting during migration, as well as the need to return to high latitudes in time to feed and replenish energy reserves. However, there is limited knowledge on the seasonality of humpback whale occurrence in the Barents Sea, limited to periods of survey and observer effort in the area. A better understanding of this

seasonality is required within the context of ongoing ecosystem changes [85]. It is unclear whether the winter feeding in coastal Norway presents a supplemental nutritional opportunity for whales or compensation for poor foraging success during the summer/autumn. Recent growth in the NEA humpback whale population [11] and food-web changes in the Barents Sea [86] may have led to a reduction in foraging success, which could be compensated for during the winter, given the high herring biomass in Norwegian waters. Increased competition due to changes in the prey base and concomitant whale population recovery has been proposed as a cause for humpback whales to feed during winter in the Pacific [85, 87]. Low prey availability on the main feeding grounds may also cause Southern Ocean humpback whales to seek out supplemental feeding opportunities [53, 74], and ecosystem changes may have caused a reduction in foraging and reproductive success in Northwest Atlantic humpback whales [88].

## Conclusion

We confirm that a successful breeding migration can take place after a winter feeding season in Norwegian fjords within the same season, and that NEA feeding grounds can be used sequentially throughout the year. Breeding humpback whale females can seemingly compensate for the long feeding season by increasing migration speed, successfully balancing it with the associated energetic costs, calf requirements and the phenology of feeding opportunities throughout the annual cycle. These findings demonstrate how individual behavioral choices can allow a whale to successfully balance energetic levels throughout the annual cycle, allowing it to adjust its movements to local prey availability. A better understanding the energetic requirements of this migration will allow researchers, managers, and policy makers to consider the needs of these top predators in ecosystem-based fisheries management, and to assess the potential impacts of increasing anthropogenic activities in the Arctic. Our results may facilitate future studies on the sensitivity of this northern population to a rapidly changing Arctic ecosystem and provide new insights in humpback whale migration ecology in the NEA and in general.

## Supporting information

**S1 File. Contains S1 and S2 Tables and S1 and S2 Figs.**
(DOCX)

**S1 Data. Contains date-time, corrected speed, movement persistence, and distance between all relocations estimated from the reconstructed path.**
(CSV)

**S2 Data. Daily latitude and longitude of the reconstructed path.**
(CSV)

## Acknowledgments

We thank Theresia Ramm, Trond Johnsen and Sune Hansen who helped with fieldwork. We are grateful for helpful comments from two anonymous reviewers and Max Czapanskiy which greatly improved this manuscript.

## Author Contributions

**Conceptualization:** Lisa Elena Kettemer, Audun H. Rikardsen, Martin Biuw, Evert Mul, Marie-Anne Blanchet.

**Data curation:** Lisa Elena Kettemer, Fredrik Broms, Evert Mul.

**Formal analysis:** Lisa Elena Kettemer.

**Funding acquisition:** Audun H. Rikardsen.

**Investigation:** Lisa Elena Kettemer, Audun H. Rikardsen.

**Methodology:** Lisa Elena Kettemer, Martin Biuw, Evert Mul, Marie-Anne Blanchet.

**Project administration:** Audun H. Rikardsen.

**Resources:** Audun H. Rikardsen.

**Supervision:** Audun H. Rikardsen, Marie-Anne Blanchet.

**Validation:** Martin Biuw.

**Visualization:** Lisa Elena Kettemer.

**Writing – original draft:** Lisa Elena Kettemer.

**Writing – review & editing:** Lisa Elena Kettemer, Audun H. Rikardsen, Martin Biuw, Fredrik Broms, Evert Mul, Marie-Anne Blanchet.

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
