## [Decision Letter · Decision Letter 0]

1 Dec 2021

PONE-D-21-34851Round-trip migration and energy budget of a breeding female humpback whale in the Northeast AtlanticPLOS ONE

Dear Dr. Kettemer,

Thank you for submitting your manuscript to PLOS ONE. After careful consideration, we feel that it has merit but does not fully meet PLOS ONE’s publication criteria as it currently stands. Therefore, we invite you to submit a revised version of the manuscript that addresses the points raised during the review process.

We look forward to receiving your revised manuscript.

Kind regards,

Vitor Hugo Rodrigues Paiva, Ph.D.

Academic Editor

PLOS ONE

Journal Requirements:

(The project was financed by the Norwegian Research Council (Whalefeast project, RFFNORD no. 282469) and LEK was funded by UiT – The Arctic University of Norway. We thank Theresia Ramm, Trond Johnsen and Sune Hansen who helped with fieldwork.)

(The project was financed by the Norwegian Research Council (Whalefeast project, RFFNORD no. 282469; https://www.forskningsradet.no) and LEK was funded by UiT – The Arctic University of Norway (https://uit.no). The funders had no role in study design, data collection and analysis, decision to publish, or preparation of the manuscript.)

5. We note that Figure 1 in your submission contain satellite images which may be copyrighted. All PLOS content is published under the Creative Commons Attribution License (CC BY 4.0), which means that the manuscript, images, and Supporting Information files will be freely available online, and any third party is permitted to access, download, copy, distribute, and use these materials in any way, even commercially, with proper attribution. For these reasons, we cannot publish previously copyrighted maps or satellite images created using proprietary data, such as Google software (Google Maps, Street View, and Earth). For more information, see our copyright guidelines: http://journals.plos.org/plosone/s/licenses-and-copyright.

A. You may seek permission from the original copyright holder of Figure 1 to publish the content specifically under the CC BY 4.0 license. 

B. If you are unable to obtain permission from the original copyright holder to publish these figures under the CC BY 4.0 license or if the copyright holder’s requirements are incompatible with the CC BY 4.0 license, please either i) remove the figure or ii) supply a replacement figure that complies with the CC BY 4.0 license. Please check copyright information on all replacement figures and update the figure caption with source information. If applicable, please specify in the figure caption text when a figure is similar but not identical to the original image and is therefore for illustrative purposes only.

6. Please include your tables as part of your main manuscript and remove the individual files. Please note that supplementary tables (should remain/ be uploaded) as separate "supporting information" files.

Reviewers' comments:

Reviewer's Responses to Questions

**Comments to the Author**

1. Is the manuscript technically sound, and do the data support the conclusions?

Reviewer #1: Yes

Reviewer #2: Yes

Reviewer #3: Yes

2. Has the statistical analysis been performed appropriately and rigorously? 

Reviewer #1: Yes

Reviewer #2: Yes

Reviewer #3: Yes

3. Have the authors made all data underlying the findings in their manuscript fully available?

Reviewer #1: Yes

Reviewer #2: Yes

Reviewer #3: Yes

4. Is the manuscript presented in an intelligible fashion and written in standard English?

Reviewer #1: Yes

Reviewer #2: Yes

Reviewer #3: Yes

5. Review Comments to the Author

Reviewer #1: This manuscript provides the first tag-derived annual record of humpback migration, which the authors interpret through the lens of migration energetics. Despite the small sample size, I believe it is a valuable addition to the literature of baleen whale migration. These data are difficult to the acquire and are invasive to the subject animal, so I do not think n=1 should hold back the paper from publication. Confirmation of secondary feeding grounds used in the same year as calving from tag data is certainly an exciting development.

The energy budget calculations are an excellent start, but I have doubts about some of the parameters used. In general, I think an uncertainty analysis is necessary to robustly interpret the results. Specifically, I wonder how uncertainty in BMR and the drag coefficient may influence these results (see comments below).

There are a few minor tweaks I ask of the authors regarding their computational methods. Most importantly, please specify the version when citing software. My only other points of confusion had to do with units used and missing entries from the bibliography. See my specific comments below for more details.

I congratulate the authors on their work and I look forward to seeing the revised version!

28: “longest [mammal] migration distance recorded”.

54-56: For capital/income breeding strategies, I would suggest citing one or more of the following foundational papers.

Drent and Daan (1980) https://doi.org/10.5253/arde.v68.p225

Jönsson (1997) https://doi.org/10.2307/3545800

Stephens et al. (2009) https://doi.org/10.1890/08-1369.1

102: What does RIB stand for? I assume rigid-hull inflatable boat, but it should be defined before first use.

103: Is ARTS an acronym or a brand?

111: Please specify the version of R.

113: Is there a reason you used a projection? In other words, why not use geodesic or great circle distances instead? I think you’d get more accurate step lengths that way.

116: For the trip package, the authors request you also cite Sumner’s dissertation (details available at https://cran.r-project.org/web/packages/trip/citation.html). Also, please specify the package version (presumably 1.8.5). I appreciate that you cited Freitas et al. (2008) for the sda function. For the sake of clarity, I would suggest stylizing function names as `function()` (i.e., fixed-width font like Courier New and followed by parentheses). In my opinion that improves readability.

119: For the foieGras package, the authors request you also cite Jonsen et al. (2020) Movement Ecology and Jonsen et al. (2019) Ecology (details at https://cran.r-project.org/web/packages/foieGras/citation.html). Also, for Jonsen and Patterson (2020), please include the DOI. It’s a Zenodo archive, so without the DOI your readers won’t be able to find it. Please specify the package version (0.7-6, I assume). For consistency, since you specified using trip::sda(), I think you should also specify which function you used from foieGras (presumably foieGras::fit_ssm()?)

130: Specify the version of adehabitatLT used. This is especially important because the paper you cited (Calenge (2006) Ecological Modelling) predates the author splitting adehabitat into multiple packages. Here again, I would suggest stylizing the function name as lavielle(). Calenge also recommends citing Lavielle (1999) https://doi.org/10.1016/S0304-4149(99)00023-X.

139: What version of marmap did you use? The latest version (1.0.6) doesn’t have a function called lc.cost(). There is a lc.dist() function – maybe that one?

148: BMR estimates are a critical component of this analysis and there is a great deal of uncertainty around the metabolic rates of very large animals (i.e., the Kleiber curve was fit to data collected from species orders of magnitude smaller than a baleen whale). I am curious whether the results presented here are robust to assumptions about metabolic rate scaling. For example, White and Seymour (2003) argued the 3/4 power law is an artifact of large terrestrial mammals’ digestive physiology and the “true” exponent is actually 2/3 (https://doi.org/10.1073/pnas.0436428100). Their allometric equation, when extrapolated to 30 000 kg, would yield much lower BMR. Conversely, Kolokotrones et al. (2010) found evidence for metabolic scaling curvature in log-log space (https://doi.org/10.1038/nature08920). Their quadratic allometric equation predicts elevated BMR in a humpback relative to Kleiber. I think the arguments in this paper would be strengthened by repeating the analysis with BMR estimates predicted by these equations. Does the finding that “pregnant females migrate faster than the energetic optimum” hold if BMR is higher than the Kleiber prediction?

155-156: You define propulsive efficiency as “how efficiently muscle work is converted to forward motion” but I think that conflates propulsive and aerobic efficiency. I believe it is more accurate to say “how efficiently mechanical work is converted to forward motion”.

163-169: Out of curiosity, how much was the contribution from ocean currents to displacement? Also, did you account for declining current speeds with depth? My (admittedly untested) assumption is current contributions would be negligible, but if your calculations indicate otherwise I would like to see that. Could you add a figure to the supplemental materials plotting corrected vs uncorrected speeds?

170: Was there a particular function you used for estimating SOG?

173-174: windR is in a living repository (as opposed to archived on CRAN or Zenodo) and there are no tagged releases, so please add “date accessed” information to the reference. Also, did you use a particular function? I would guess windSupport().

196-198: I take it these distances were calculated using the projected coordinates, correct? How different are they if you use geodesic distances? At these spatial scales I think it’s possible the errors could add up to a substantial amount.

226-227: I think these units are wrong. Watts are a measure of power (energy per unit time) and I believe you summed power over time, so this should be in Joules.

279-293: Your results about the timing of calving are fascinating! Very excited to see this recorded with tag data.

300: This is largely why I’m curious about how much ocean currents contributed to swimming speeds. How much could the other studies be off by?

339-343: I find this particularly interesting, as it means time, not energy, is the important currency. To me, this suggests rorqual whales are such efficient foragers that they are essentially food unconstrained during the feeding season. What does it matter if you burn more calories during migration if you’re a feeding powerhouse for a few months of the year? Relative to their body size, I think rorqual whales may actually have very fast-paced life histories. Perhaps it’s out of the scope of the present manuscript, but I’m curious what the authors think their results say about pace-of-life at extreme body sizes.

Table 1: You cite references 66 and 67, but your bibliography only goes up to 63. Also, I don’t see 64 or 65 cited anywhere. Regarding drag, a recent study (Gough et al., 2021) using tag data derived a drag coefficient five times greater than the value used in the model (https://doi.org/10.1242/jeb.237586). Granted, that analysis looked at a finer temporal scale than migration, but it does raise questions about the model’s parameterization. Based on equation 2, ECOT is directly proportional to CD. Do the authors’ results and interpretations change if ECOT is 5x greater?

Table 2: The units in this table are very confusing to me. E KM-1 should be in units of J km-1 maybe? It’s not W, which is energy per unit time (not distance). Similarly, I can’t understand E DAY-1 (W), since W is J s-1, not J day-1. I believe ECOT, BMR, and ETOTAL should be in J, not W, since you integrated over time (if I understand correctly). I think BMR should be “Total basal metabolic expenditure” or something like that, since after integrating you’re no longer talking about a rate.

Figure 1: I really like these maps. Just a couple comments. 1) I would change the legend title of the inset from “g” to “” to better match the text. 2) I think the inset legend would be cleaner if you expanded the limits to include 0 and 1. 3) In the main map, you overlay red boxes on green points. It’s unclear what the red boxes represent. I assume the West Indies red box delineates the inset, but I’m unsure what the four northern boxes are. Also, people with color blindness will have a hard time interpreting red on green, so please choose a colorblind friendly combination of colors. Lastly, you cite (68) but the bibliography only goes up to 63.

Figure 3: Great figure. My only request is you make the movement persistence gradient in this figure match the gradient in the figure 1 inset, for consistency.

Figure 4: This is a helpful figure, but I’m unclear how you calculated the first and third quartiles. If the dotted lines represent Q1 and Q3 for each segment, then shouldn’t there be the same number of points below the Q1 line and above the Q3 lines? For example, the second segment appears to have 9 points below Q1 and 19 above Q3. The next segment has only two points above Q3 and 13 below Q1. How did you come up with those quartiles? Also, and this is really nitpicky, the interquartile range is (technically speaking) the difference between Q3 and Q1 (i.e., it’s a single number). I think it would be more accurate to say “dotted lines represent the first and third quartiles”.

Reviewer #2: This study present a detailed description of a full migration cycle for a breeding humpback whale, estimates the energetic cost of the long migration and compares the results to previous work. I believe that this research will be of interest to many researchers, especially those studying large whales by using biologging techniques. I found the work very interesting, and thought that overall the paper was clearly written and easy to follow/understand.

Below I have provided few comments and suggestions to improve the manuscript.

Line 71-72: “They may therefore face higher energetic constraints compared to whales in the western Atlantic” – you should provide more context for this comparison by mentioning the breeding and feeding grounds between which the whales in the western Atlantic migrate, as most readers might not know what they are and can’t form a mental image of how and why the NE Atlantic humpbacks would face a higher energetic constraint compared to the W Atlantic whales.

Line 76: “Norwegian spring spawning (NSS) herring” – NSS acronym not necessary as not mentioned again in the manuscript

Lines 225-227: “We estimated the total cost of the round-trip migration (from coastal Norway to the Barents Sea) to be 2 681 937 Watts (W), where 51% (1 365 516 W) constitutes cost of transport and the rest represents maintenance metabolism (1 316 420 W, Table 2)” -- I think you should briefly mention (probably in the Discussion) the uncertainty around this exact W value provided, due to the uncertainty of the various parameters (e.g., for some of the parameters there is no exact value for humpback whales, plus the value of mass used in the model may not have been the exact size of the individual whale etc. etc.).

Also, because we know for sure that this individual was gestating and then lactating (lactation being hugely energetically demanding), I think it should be mentioned somewhere how the exact W value will be an underestimate of the TRUE energy amount (W) that the animal used (although 51% of this estimate without lactation was transport costs, 51% of the TRUE round-trip cost probably wasn’t travel….). And a follow up to that point, did you consider adding lactation cost to the model (unsure if that exists for humpback whales…) - how would that have affected the calculations and results? It could be worth addressing why lactation cost (and gestation cost) was not added into the calculations, as most people will probably ask about that. But I think the discussions and comparisons around the cost of transport are fine as the (only) key parameter there is the speed of travel.

Lines 239-241: “Earlier studies on humpback whales in other regions have only mapped their migrations one way” - it could be worth mentioning here that this isn’t necessarily due to lack of trying but due to lack of data as the tags haven’t lasted long enough to capture full migrations (leading to an optional discussion point: developments in the tags, perhaps the newer tags such as used here are better in some ways?).

Lines 260-261: “most individuals likely travel directly from the main foraging grounds to breeding grounds” – given that you calculated the most direct route and showed that the tagged individual did not take the straightest possible route when migrating, I think there is a great chance to add some broader (brief) discussion on ‘why don’t animals just take THE shortest route possible’, and in the case of this whale what might be the reasons for not taking the most direct/shortest route (e.g. is it because they learn the route from their mother…?)

Lines 266-268: “The departure timing from Norway on February 7th is slightly later than that of other whales feeding in coastal Norway, as they generally depart between December and late January (16).” – I think you should expand here about this ‘late departure’ and why that might be – the hierarchical migratory timing of humpbacks. You already mention this a bit in the introduction (lines 85-86: “While pregnant humpback whales commonly maximize the time spent on feeding grounds (17–19)”).

Lines 289-290: ‘humpback whale calving events in the Southern Ocean have been reported as far south as 15 -20°S, outside of the main breeding grounds” – when you say “calving events in the SO outside of the main breeding grounds”, I wonder what you mean exactly… I think this sentence needs some edits as it doesn’t sound quite right. The SO (starting at 60S) is the feeding ground for southern hemisphere humpbacks. Also, on the ‘as far south as 15-20S’ – in New Caledonia a key breeding site is the south lagoon which is around 22S, and therefore further south than the ‘as far south as 15-20S’. The second half of this sentence on lines 291-292 “supporting the idea that births can occur over a large geographic area and within a 4- 6-month window” is really good, I just think that the first half of the sentence needs some adjusting and clarifying.

Line 300: “none of these studies accounted for ocean currents.” – Few thoughts around incorporating ocean currents in the speed calculations:

1. I think you might want to briefly touch on any caveats with the data you used (e.g. missing data for some time periods (for what % of locations during active migration was current data not available? How might that affect the results?), ocean current data probably reflects surface waters and although humpbacks do hang around near the surface quite a lot they obviously wouldn’t be affected the same way if they dive deeper?).

2. You make the point that other studies didn’t account for ocean currents. What is the migration speed in your data if you don’t account for ocean currents? It would be interesting to see what your results would be if you hadn't corrected for currents and would they then have been more similar to the previous studies?

Lines 319-320: “faster swim speeds can lead to a loss of milk and increased energetic demands of the calf,” – I think you need to briefly mention how mothers and calves swim in echelon formation which allows the calf to slip stream and not use much energy (though on the flip side that changes the drag the mother experiences – you should acknowledge this as well as this change in drag was not included in the model).

Lines 322-338: This is a really interesting comparison! So the energetic cost of the northward migration for this one animal was twice the average cost reported for females migrating with calves from the Southern Ocean to Australia – but – the distance for this one animal was also twice as long, so technically that would mean the transport cost for a comparable distance (e.g. for every 1000km travelled) was actually the same. However, this one animal travelled a lot faster. I wonder what effect the inclusion of ocean currents in your model had here… Either way, it is an interesting chance to compare model results, and you do a nice job discussing the tradeoffs between travelling faster (and using more energy) vs having more time on the feeding grounds. I wonder if there has been or will be any work in the Barent Sea/Norway area to try to estimate the energy gained by humpback whales during the feeding season (as the annual energy budget will be a function of both energy used during the year and energy gained during the year).

And now that I have read that section of the discussion, going back to the abstract lines 37-39: “The estimated energetic cost of this migration was substantially higher than the energetic costs of other humpback whale migrations, resulting from the long migration distance and fast migration speed (1.6 ms-1).” – you might want to be mindful of the wording, because if you are comparing the energetic costs of a full migratory cycle to only a half a migratory cycle then of course it will be higher. And based on the discussion section, if we just double the southern hemisphere half a migratory cycle cost then it actually ends up being about the same, i.e., no substantial difference between this migration cost and those in the southern hemisphere. Maybe just adjust the wording a bit so that you aren’t making claims that you don’t mean to/that aren’t backed up by your results.

Figure 1: Because figures should be fully standalone from the text, I think you need to explain movement persistence in the legend a little bit for those readers who aren’t familiar with state-space models so that they don’t have to search for the information in the main text while looking at the figure. I would suggest the following: the ‘g’ in the scale should be mentioned in the legend e.g., by saying “… movement persistence (g)…” followed by something along the lines “ ..lighter blue colours indicating directional travel, and darker blue colours indicating meandering movement..”

Also, it looks like in the insert the line is on top of the blue points, causing it to be a little hard to make out the difference between the lighter and darker shades of blue – I would suggest placing the line under the points so that the colour differences are bit clearer.

Figure 2: maybe provide exact dates for the photos?

Table S.2: I don’t think it would actually be of much relevance for your work, but some variability in migratory speed between the individual humpback whales in the southern hemisphere are provided in https://doi.org/10.1016/j.ecolind.2018.02.030 (although calculations were done at a 12 hour time step).

Really interesting work!

Reviewer #3: The authors attached a satellite tag to a humpback whale and successfully recorded its full, round trip, long migration between feeding ground and breeding ground. Importantly, the whale calved during the migration (based on photo ID), providing important insights into the breeding biology of this species. The paper is well written and presented. Although sample size is just one, I recommend that this paper be published in Plos One.

Minor comments

Title: energy budget => energetic cost?

L117 pre-filtered => filtered?

L174 current corrected swim speed => estimated swim speed through water?

L226, Table 2 and elsewhere. Please pay attention to unit. Total cost should be represented by Joule rather than Watt if you are interested in a total cost over time. Energetic costs per km (or per day) in Watt (Table 2) don't make sense.

Table 1. Lambda is <1, which means that animals incur less drag during active swimming than gliding. This seems to be a very unusual case (see, for example, Weihs 1973 J. Mar. Res.). I cannot find the cited reference (37). Please provide a regular article for the citation here. Also, papers with reference no. of >63 are not listed.

6. PLOS authors have the option to publish the peer review history of their article (what does this mean?). If published, this will include your full peer review and any attached files.

Reviewer #1: **Yes: **Max Czapanskiy

Reviewer #2: No

Reviewer #3: No

---

## [Author Response · Author response to Decision Letter 0]

9 Mar 2022

Response to Reviewers has been uploaded as a separate PDF file.

---

## [Decision Letter · Decision Letter 1]

19 Apr 2022

PONE-D-21-34851R1Round-trip migration and energy budget of a breeding female humpback whale in the Northeast AtlanticPLOS ONE

Dear Dr. Kettemer,

Thank you for submitting your manuscript to PLOS ONE. After careful consideration, we feel that it has merit but does not fully meet PLOS ONE’s publication criteria as it currently stands. Therefore, we invite you to submit a revised version of the manuscript that addresses the points raised during the review process.

We look forward to receiving your revised manuscript.

Kind regards,

Vitor Hugo Rodrigues Paiva, Ph.D.

Academic Editor

PLOS ONE

Journal Requirements:

Reviewers' comments:

Reviewer's Responses to Questions

**Comments to the Author**

1. If the authors have adequately addressed your comments raised in a previous round of review and you feel that this manuscript is now acceptable for publication, you may indicate that here to bypass the “Comments to the Author” section, enter your conflict of interest statement in the “Confidential to Editor” section, and submit your "Accept" recommendation.

Reviewer #2: (No Response)

Reviewer #3: All comments have been addressed

2. Is the manuscript technically sound, and do the data support the conclusions?

Reviewer #2: Yes

Reviewer #3: Yes

3. Has the statistical analysis been performed appropriately and rigorously? 

Reviewer #2: Yes

Reviewer #3: Yes

4. Have the authors made all data underlying the findings in their manuscript fully available?

Reviewer #2: Yes

Reviewer #3: Yes

5. Is the manuscript presented in an intelligible fashion and written in standard English?

Reviewer #2: Yes

Reviewer #3: Yes

6. Review Comments to the Author

Reviewer #2: This manuscript reports on a round-trip migration and energy budget of a breeding female humpback whale in the Northeast Atlantic. I believe that the authors have sufficiently addressed the comments made by the reviewers.

There are a few minor edits that still need to be made for the manuscript to be fully acceptable for publication.

Abstract (line 38): consider editing the sentence to say “…presumably in response to the calf’s needs after its birth.”

Methods: Sometimes there seems to be an extra space between a function name and the brackets.

There are some issues with the figure legends:

Lines 111 and 120 both are legends for ‘Figure 1’, and therefore Line 211 Fig 2 legend should in fact be Fig 3, and line 245 Fig 3 legend should in fact be Fig 4.

Line 111 for Fig 1: the legend states that “Tagging location is indicated by a red triangle and the putative location of birth is indicated by a red star.” But these are in fact denoted by yellow/orange triangle and square. Furthermore, the legend text about the move persistence states that “lighter blue colors indicating directional travel, and darker blue colors indicating meandering movement.” whereas the color legend ranges from yellow to dark blue.

Line 255: “…was only detected after the pair transited to Navidad Bank on May 1st (Fig. 2)”. – I think this line should probably be referring to Fig 1 or Fig 3, as Fig 2 is pictures of the whales…?

Line 266-267: “We also present a theoretical cost at lower migration speeds (as reported in previous studies, 0.9 m s-1 and 1.1 m s-1) in S1 Table in S1 Supporting information.” – I this line should be referring to Table S2?

There has likely been some issue in submitting Figure S2 as nothing is plotted in the image.

Lines 376-378: “The fast migration speed and long distance resulted in a high energetic cost of this migration, compared to e.g., lactating females migrating 0.6 ms-1 slower and ~ half the distance from the Southern Ocean to Australia [7].” -- in [7] the migration direction was in fact from Australia/north of New Zealand towards Southern Ocean, not from the Southern Ocean to Australia.

Reviewer #3: (No Response)

7. PLOS authors have the option to publish the peer review history of their article (what does this mean?). If published, this will include your full peer review and any attached files.

Reviewer #2: No

Reviewer #3: No

---

## [Author Response · Author response to Decision Letter 1]

26 Apr 2022

Response to Reviewers has also been added as a PDF file.

Abstract (line 38): consider editing the sentence to say “...presumably in response to the calf’s needs after its birth.”

Implemented.

Methods: Sometimes there seems to be an extra space between a function name and the brackets.

Corrected.

There are some issues with the figure legends:

Lines 111 and 120 both are legends for ‘Figure 1’, and therefore Line 211 Fig 2 legend should in fact be Fig 3, and line 245 Fig 3 legend should in fact be Fig 4.

Line 111 for Fig 1: the legend states that “Tagging location is indicated by a red triangle and the putative location of birth is indicated by a red star.” But these are in fact denoted by yellow/orange triangle and square. Furthermore, the legend text about the move persistence states that “lighter blue colors indicating directional travel, and darker blue colors indicating meandering movement.” whereas the color legend ranges from yellow to dark blue.

Corrected.

Line 255: “...was only detected after the pair transited to Navidad Bank on May 1st (Fig. 2)”. – I think this line should probably be referring to Fig 1 or Fig 3, as Fig 2 is pictures of the whales...?

Corrected.

Line 266-267: “We also present a theoretical cost at lower migration speeds (as reported in previous studies, 0.9 m s-1 and 1.1 m s-1) in S1 Table in S1 Supporting information.” – I this line should be referring to Table S2?

Corrected.

There has likely been some issue in submitting Figure S2 as nothing is plotted in the image.

In our version of the pdf compiled submission material this figure is visible. We will check whether this problem persists in the following process.

Lines 376-378: “The fast migration speed and long distance resulted in a high energetic cost of this migration, compared to e.g., lactating females migrating 0.6 ms-1 slower and ~ half the distance from the Southern Ocean to Australia [7].” -- in [7] the migration direction was in fact from Australia/north of New Zealand towards Southern Ocean, not from the Southern Ocean to Australia.

Corrected.

---

## [Editor Report · Decision Letter 2]

28 Apr 2022

Round-trip migration and energy budget of a breeding female humpback whale in the Northeast Atlantic

PONE-D-21-34851R2

Dear Dr. Kettemer,

We’re pleased to inform you that your manuscript has been judged scientifically suitable for publication and will be formally accepted for publication once it meets all outstanding technical requirements.

Kind regards,

Vitor Hugo Rodrigues Paiva, Ph.D.

Academic Editor

PLOS ONE
---

## [Editor Report · Acceptance letter]

18 May 2022

PONE-D-21-34851R2 

Round-trip migration and energy budget of a breeding female humpback whale in the Northeast Atlantic 

Dear Dr. Kettemer:

I'm pleased to inform you that your manuscript has been deemed suitable for publication in PLOS ONE. Congratulations! Your manuscript is now with our production department. 

Kind regards, 

on behalf of

Dr. Vitor Hugo Rodrigues Paiva 

Academic Editor

PLOS ONE